# NAS-Bench-360: Benchmarking
# Neural Architecture Search on Diverse Tasks

**Renbo Tu**[*]
University of Toronto
renbo.tu@mail.utoronto.ca

**Nicholas Roberts**[*]
University of Wisconsin
nick11roberts@cs.wisc.edu

**Mikhail Khodak**
Carnegie Mellon University
khodak@cmu.edu

**Junhong Shen**
Carnegie Mellon University
junhongs@andrew.cmu.edu

**Frederic Sala**
University of Wisconsin
fsala@wisc.edu

**Ameet Talwalkar**
Carnegie Mellon University
talwalkar@cmu.edu

## Abstract

Most existing neural architecture search (NAS) benchmarks and algorithms prioritize well-studied tasks, e.g. image classification on CIFAR or ImageNet. This makes the performance of NAS approaches in more diverse areas poorly understood. In this paper, we present **NAS-Bench-360**, a benchmark suite to evaluate methods on domains beyond those traditionally studied in architecture search, and use it to address the following question: *do state-of-the-art NAS methods perform well on diverse tasks?* To construct the benchmark, we curate ten tasks spanning a diverse array of application domains, dataset sizes, problem dimensionalities, and learning objectives. Each new task is carefully chosen to interoperate with modern convolutional neural network (CNN) search methods while being far-afield from their original development domain. To speed up and reduce the cost of NAS research, for two of the tasks we release the precomputed performance of 15,625 architectures comprising a standard CNN search space. Experimentally, we show the need for more robust NAS evaluation of the kind NAS-Bench-360 enables by showing that several modern NAS procedures perform inconsistently across the ten tasks, with many catastrophically poor results. We also demonstrate how our benchmark and its associated precomputed results will enable future scientific discoveries by testing whether several recent hypotheses promoted in the NAS literature hold on diverse tasks. NAS-Bench-360 is hosted at https://nb360.ml.cmu.edu/.

## 1 Introduction

Neural architecture search (NAS) aims to automate the design of deep neural networks, ensuring performance on par with hand-crafted architectures while reducing human labor devoted to tedious architecture tuning [20]. With the growing number of application areas of ML, and thus of use-cases for automating it, NAS has experienced an intense amount of study in well-established machine learning domains, with significant progress in search space design [63, 42, 6], search efficiency [46], and search algorithms [55, 37, 53]. Notably, the field has largely been dominated by methods designed for and evaluated on benchmarks in computer vision [42, 56, 17], yet the use of NAS techniques may

---

[*]Equal contribution

36th Conference on Neural Information Processing Systems (NeurIPS 2022) Track on Datasets and Benchmarks.

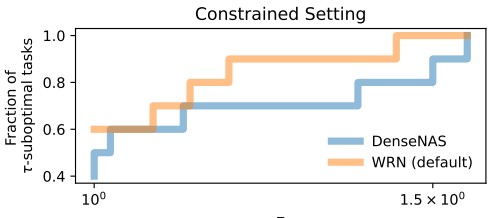 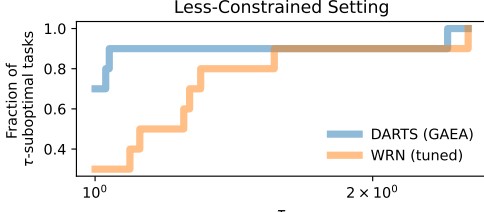

Figure 1: Performance profiles on NAS-Bench-360 comparing NAS methods (blue) to a fixed CNN (orange), specifically a Wide ResNet (WRN) [57]. Resource-constrained practitioners might be better off not using NAS (left), while less constrained practitioners can still benefit (right). The y-axis is the fraction of tasks on which error is within a factor $\tau$ of the optimal method, i.e. higher is better.

be especially impactful in under-explored or under-resourced domains where less is known about useful architecture design patterns. There have been a few recent efforts to diversify these benchmarks to settings such as vision-based transfer learning [18] and speech and language processing [43, 33]; however, evaluating NAS methods on such well-studied tasks using traditional CNN search spaces does not give a good indication of their utility on more far-afield applications, which have often necessitated the design of custom neural operations [10, 40].

We make progress towards studying NAS on more diverse tasks by introducing a suite of benchmark datasets drawn from various data domains that we collectively call **NAS-Bench-360**. This benchmark consists of an organized setup of ten suitable datasets that represent diverse application domains, dataset sizes, problem dimensionalities, and learning objectives. We also include a standard image classification task as a baseline point of comparison, as many new methods continue to be designed for that setting. Note that the core component of NAS-Bench-360 is *not* a typical NAS benchmark, which often involves precomputing all architectures in some fixed search space. In contrast, our contribution is explicitly intended to be agnostic of the search space being used, as different search spaces may work well for different tasks. Thus NAS-Bench-360 is a task-oriented NAS benchmark with the intended use-case of evaluating NAS method and search space pairs on a wide variety of domains. However, to aid research, three of our tasks—for two of which we contribute the precompute—do come with trained architectures from the NAS-Bench-201 search space [17].

Experimentally, we demonstrate the usefulness of NAS-Bench-360 by performing a set of analyses evaluating whether the success of NAS in computer vision is indicative of strong performance on the much broader set of problems to which NAS can be applied. Specifically, we report performance comparisons between NAS methods, investigate the validity of existing NAS hypotheses made solely on computer vision tasks, and extend an existing analysis of zero-cost proxies already-enabled by our benchmark [52]. From these analyses, we arrive at the following conclusions:

- Resource-constrained practitioners may be better of using a fixed CNN rather than NAS (Figure 1).
- NAS-Bench-201 analyses on computer vision tasks do not generalize to diverse tasks.
- Zero-cost proxies perform inconsistently on diverse tasks, corroborating earlier findings [52].

We have released all datasets, experiment code, precomputed models, seeds, and environments used in our experiments.[1] Releasing our code, random seeds, and environments in the form of Docker containers assures reproducibility of all experimental results presented in this work and encourages the same level of reproducibility for future research performed using NAS-Bench-360.

## 2 Related Work

Benchmarks have been critical to the development of NAS in recent years. This includes standard evaluation datasets and protocols, of which the most popular are the CIFAR-10 and ImageNet routines used by DARTS [42]. Another important type of benchmark has been tabular benchmarks such as NAS-Bench-101 [56], NAS-Bench-201 [17], NAS-Bench-1Shot1 [58], and TransNAS-Bench-101 [19]; these benchmarks exhaustively evaluate all architectures in their search spaces, which is made computationally feasible by defining simple searched cells. Consequently, they are less

---

[1] https://github.com/rtu715/NAS-Bench-360

expressive than the DARTS cell [42], often regarded as the most powerful search space in the cell-based regime. Notably, the full NAS-Bench-360 benchmark is *not* intended to be a tabular benchmark, i.e. we do *not* evaluate every architecture from a fixed search space on all ten of our tasks; instead, the focus is on the organization of a suite of tasks for assessing both NAS algorithms and search spaces, which would necessarily be restricted by fixing a search space for a tabular benchmark. Pre-computing on an expansive search space such as DARTS, with $10^{18}$ possible architectures, is computationally intractable. Architectures found on lesser search spaces are most likely suboptimal: a vanilla Wide ResNet (WRN) outperforms all networks in the NAS-Bench-201 search space on CIFAR-100. Nonetheless, we find that including precompute results for all of NAS-Bench-201 on two of our tasks is useful in evaluating various claims in the NAS literature centered on computer vision tasks.

While NAS methods and benchmarks have generally been focused on computer vision, recent work such as AutoML-Zero [48] and XD-operations [49] has started moving towards a more generically applicable set of tools for AutoML. However, even more recent benchmarks that do go beyond the most popular vision datasets have continued to focus on well-studied tasks, including vision-based transfer learning [18], speech recognition [43], and natural language processing [33]. We aim to go beyond such areas to evaluate the potential of NAS to automate the application of ML in truly under-explored domains. One analogous work to ours in the field of meta-learning is the Meta-Dataset benchmark of few-shot tasks [50], which similarly aimed to establish a wide-ranging set of evaluations for that field. For our inclusion of diverse tasks, we title our benchmark NAS-Bench-360 to resemble the idea of a 360-degree camera that covers all possible directions.

# 3 NAS-Bench-360: A Suite of Diverse and Practical Tasks

In this section, we introduce the NAS setting targeted by our benchmark, our motivation for organizing a new set of diverse tasks as a NAS evaluation suite, and our task-selection methodology. We report evaluations of specific algorithms on this new benchmark in the next section.

## 3.1 Neural Architecture Search: Problem Formulation and Baselines

For completeness and clarity, we first formally discuss the architecture search problem itself, starting with the extended hypothesis class formulation [37]. Here the goal is to use a dataset of points $x \in \mathcal{X}$ to find parameters $\mathbf{w} \in \mathcal{W}$ and $a \in \mathcal{A}$ of a parameterized function $f_{\mathbf{w},a} : \mathcal{X} \mapsto \mathbb{R}_{\geq 0}$ that minimize the expectation $\mathbb{E}_{x \sim \mathcal{D}} f_{\mathbf{w},a}(x)$ for some test distribution $\mathcal{D}$ over $\mathcal{X}$; here $\mathcal{X}$ is the input space, $\mathcal{W}$ is the space of model weights, and $\mathcal{A}$ is the set of architectures. For generality, we do not require the training points to be drawn from $\mathcal{D}$ to allow for domain adaptation, as is the case for one of our tasks, and we do not require the loss to be supervised. Note also that the goal here does not depend on computational or memory efficiency, which we do not focus on in our evaluations; our restriction is only that the entire pipeline can be run on an NVIDIA V100 GPU.

Notably, this formulation makes no distinction between the model weights $\mathbf{w}$ and architectures $a$, treating both as parameters of a larger model. Indeed, the goal of NAS may be seen as similar to model design, except now we include the design of an (often discrete) *architecture space* $\mathcal{A}$ such that it is easy to find an architecture $a \in \mathcal{A}$ and model weights $\mathbf{w} \in \mathcal{W}$ whose test loss $\mathbb{E}_{\mathcal{D}} f_{\mathbf{w},a}$ is low using a search algorithm. This can be done in a one-shot manner—simultaneously optimizing $a$ and $\mathbf{w}$—or using the standard approach of first finding an architecture $a$ and then keeping it fixed while training model weights $\mathbf{w}$ using a pre-specified algorithm such as stochastic gradient descent (SGD). This formulation divides NAS algorithms into two camps: one-shot, weight-sharing methods and non-weight-sharing ones such as random search, which operate by repeatedly sampling architectures and evaluating them. The formulation also includes non-NAS methods by allowing the architecture search space to be a singleton. When the sole architecture is a standard and common network such as WRN [57], this yields a natural baseline with an algorithm searching for training hyperparameters, not architectures. For our empirical investigation, we compare the performance of state-of-the-art NAS approaches against that of the three baselines: WRN, PerceiverIO [30], and XGBoost [7].

## 3.2 Task Selection: Motivation and Methodology

Curating a diverse, practical set of tasks for the study of NAS is our primary motivation behind this work. We observe that past NAS benchmarks focused on creating larger search spaces and more

Table 1: Task metadata for NAS-Bench-360. Metrics are standardized such that lower is better.

| Task name | Size | Dim. | Type | Learning objective | Metric | New to NAS? |
|-----------|------|------|------|--------------------|--------|-------------|
| CIFAR-100 [34] | 60K | 2D | Point | Classify natural images into 100 classes | 0-1 error | no, widely used |
| Spherical [10] | 60K | 2D | Point | Classify spherically projected images into 100 classes | 0-1 error | ✓ |
| NinaPro [4] | 3956 | 2D | Point | Classify sEMG signals into 18 classes of hand gestures | 0-1 error | ✓ |
| FSD50K [24] | 51K | 2D | Point (multi-label) | Classify sound events in log-mel spectrograms with 200 labels | $1 - mAP$ | ✓ |
| Darcy Flow [40] | 1100 | 2D | Dense | Predict the final state of a fluid from its initial conditions | relative $\ell_2$ | no, used in [49] |
| PSICOV [3] | 3606 | 2D | Dense | Predict pairwise distances between residuals from pairwise sequence features | $MAE_8$ | no, used in [49] |
| Cosmic [59] | 5250 | 2D | Dense | Predict probabilistic maps to identify cosmic rays in telescope images | 1 - AUROC | ✓ |
| ECG [9] | 330K | 1D | Point | Detecting atrial cardiac disease from ECG recordings | $1 - F1$ | ✓ |
| Satellite [45] | 1M | 1D | Point | Classify satellite image pixel time series into 24 land cover types | 0-1 error | ✓ |
| DeepSEA [11] | 250K | 1D | Point (multi-label) | Predicting chromatin and binding states of RNA sequences | $1 - AUROC$ | no, used in [60, 61] |

sophisticated search methods for neural networks. However, the utility of these search spaces and methods are only evaluated on canonical computer vision datasets. On a broader range of problems, whether these new methods can improve upon simple baselines remains an open question. This calls for the introduction of new datasets lest NAS research overfits to the biases of CIFAR-10 and ImageNet. By identifying these possible biases, future directions in NAS research can be better primed to suit the needs of practitioners and to increase the deployment of NAS.

Summarized in Table 1, NAS-Bench-360 consists of problems that are conducive to processing by convolutional neural networks, which includes a trove of applications associated with spatial and temporal data, spanning single and multiple dimensions. Most current NAS methods are not implemented to search for other types of architectures to process tabular data and graph data. Therefore, we have set this scope for our investigation. During the selection of tasks, diversity is our primary consideration. We define the following axes of diversity to govern our task-filtering process: the first is problem dimensionality, including both 2D with matrix inputs and 1D with sequence inputs; the second is dataset size, for which our selection spans the scale from 1,000 to 1,000,000; the third is problem type , divisible into tasks requiring a singular prediction (point prediction) and multiple predictions (dense prediction); fourth and finally, diversity is achieved through selecting tasks from various learning objectives from applications of deep learning, where introducing NAS could improve upon the performance of handcrafted neural networks.

In lieu of providing raw data, we perform data pre-processing locally and store the processed data on a public Amazon Web Services S3 data bucket with download links available on our website. Our data treatment largely follows the procedure defined by the researchers who provided them. This enhances reproducibility by ensuring the uniformity of input data for different pipelines. Additional information about the datasets, pre-processing, and augmentation steps are described in the Appendix.

Table 2: Performance of NAS and baselines across NAS-Bench-360. Methods are divided into efficient methods (e.g. DenseNAS and fixed WRN) that take 1-10 GPU-hours, more expensive methods (e.g. DARTS and tuned WRN) that take 10-100+ GPU-hours, and specialized methods (Auto-DL and AMBER). All results are averages of three random seeds, and lower is better for all metrics. The best performing method is shown in **bold** and the best non-expert-designed method is underlined.

| Search space | Search algorithm | CIFAR-100 | Spherical | Darcy Flow | PSICOV | Cosmic |
|---|---|---|---|---|---|---|
| WRN | default | 23.35±0.05 | 85.77±0.71 | 0.073±0.001 | 3.84±0.05 | 0.245±0.02 |
| DenseNAS | random | 25.49±0.41 | 71.23±1.65 | 0.071±0.006 | 3.70±0.06 | 0.309±0.04 |
| DenseNAS | original | 25.98±0.38 | 72.99±0.95 | 0.100±0.010 | 3.84±0.15 | 0.383±0.04 |
| Perceiver IO | default | 70.04±0.44 | 82.57±0.19 | 0.240±0.010 | 8.06±0.06 | 0.485±0.01 |
| XGBoost | default | 84.83±4.15 | 96.92±0.02 | 0.085±0.000 | n/a* | 0.232±0.00 |
| WRN | ASHA | 23.39±0.01 | 75.46±0.40 | 0.066±0.000 | 3.84±0.05 | 0.251±0.02 |
| DARTS | GAEA | 24.02±1.92 | **48.23±2.87** | 0.026±0.001 | **2.94±0.13** | 0.229±0.04 |
| Auto-DL | DARTS | n/a | n/a | 0.049±0.005 | 6.73±0.73 | 0.495±0.00 |
| Expert | default | **19.39±0.20** | 67.41±0.76 | **0.008±0.001** | 3.35±0.14 | **0.127±0.01** |

| Search space | Search algorithm | NinaPro | FSD50K | ECG | Satellite | DeepSEA |
|---|---|---|---|---|---|---|
| WRN | default | **6.78±0.26** | 0.92±0.001 | 0.43±0.01 | 15.49±0.03 | 0.40±0.001 |
| DenseNAS | random | 8.45±0.56 | **0.60±0.001** | 0.42±0.01 | 13.91±0.13 | 0.40±0.001 |
| DenseNAS | original | 10.17±1.31 | 0.64±0.002 | 0.40±0.01 | 13.81±0.69 | 0.40±0.001 |
| Perceiver IO | default | 22.22±1.80 | 0.72±0.002 | 0.66±0.01 | 15.93±0.08 | 0.38±0.004 |
| XGBoost | default | 21.90±0.70 | 0.98±0.002 | 0.56±0.00 | 36.36±0.02 | 0.50±0.000 |
| WRN | ASHA | 7.34±0.76 | 0.91±0.030 | 0.43±0.01 | 15.84±0.52 | 0.41±0.002 |
| DARTS | GAEA | 17.67±1.39 | 0.94±0.020 | 0.34±0.01 | **12.51±0.24** | 0.36±0.020 |
| AMBER | ENAS | n/a | n/a | 0.33±0.02 | 12.97±0.07 | 0.32±0.010 |
| Expert | default | 8.73±0.90 | 0.62±0.004 | **0.28±0.00** | 19.80±0.00 | **0.30±0.024** |

* did not fit on a single V100 GPU.

# 4 Experimental design

Having detailed our construction of NAS-Bench-360, in this section we will establish the experimental setup for our analyses in the following section, which demonstrates the usefulness of NAS-Bench-360 for evaluating NAS methods on diverse tasks. We first specify the NAS methods and baselines we compare, followed by the details of the experimental setup and intended use of the benchmark. Finally, we provide details of the precomputed NAS-Bench-201 search space for two representative diverse tasks from NAS-Bench-360: NinaPro and Darcy Flow.

## 4.1 Baselines and Search Procedures

Our initial experiments follow two practitioners with different resource settings: one with enough compute to tune a WRN (less-constrained) and another who can only train it once with the default hyperparameters (constrained). Given these two scenarios, we compare against NAS methods that each practitioner would be able to run. In both cases, we focus on two well-known search paradigms: cell-based NAS (using DARTS [42]) and macro NAS (using DenseNAS [22]). We further compare these approaches to two customized NAS methods: Auto-DeepLab [41] for 2D dense prediction and AMBER [61] for 1D prediction, as well as general-purpose baselines: Perceiver IO [30] and XGBoost [7]. Additional details are provided in the Appendix.

## 4.2 Experimental Setup

Below we discuss the main reporting details of our empirical evaluation.

Table 3: Median rank and performance improvement over WRN across NAS-Bench-360.

| Search space Search algorithm | WRN default | DenseNAS original | DenseNAS random | WRN ASHA | DARTS GAEA | Auto-DL DARTS | AMBER ENAS |
|---|---|---|---|---|---|---|---|
| Median rank | 4.0 | 4.0 | 4.0 | 3.5 | 1.5 | 6.0[†] | 1.0[†] |
| % better than WRN* | 0.0% | 2.53% | 0.0% | 0.1% | 14.6% | -75.3%[†] | 20.0%[†] |

* relative improvement over the default (untuned) WRN baseline
[†] metric computed only on the subset of three tasks on which the method was evaluated

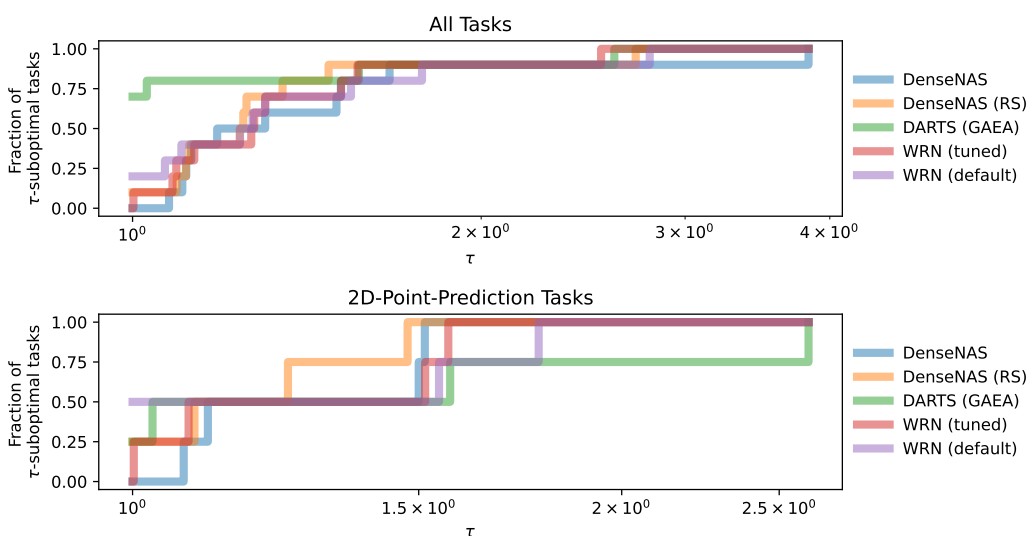

Figure 2: In our investigation of modern methods on NAS-Bench-360, we find that methods like GAEA DARTS can be strong in aggregate, as shown in the performance profiles on all tasks (top), but worse on salient subsets such as 2D point tasks (bottom). The y-axis is the fraction of tasks on which error is within a factor $\tau$ of the optimal method, i.e. higher is better.

- **Hyperparameter tuning:** As detailed in the Appendix, we use the same hyperparameter ranges across all tasks to tune WRN. We use ASHA [36] to search over these hyperparameters and give it a budget on each task that matches the total search and retraining budget of DARTS (GAEA).
- **Aggregation metrics:** To aggregate results across tasks, we use the median rank of each method and its performance improvement over WRN for direct comparison via a singe number, as demonstrated in Table 3. However, since these metrics can be sensitive to small differences in performance, we also employ performance profiles [14] to mitigate that effect while still accounting for outliers. As described in Figure 1, these curves denote for each $\tau$ the fraction of tasks on which a method is no worse than a $\tau$-factor from the optimal. Concretely, we plot $\rho_s(\tau) = \frac{1}{|\mathcal{P}|} \left| \left\{ p \in \mathcal{P} : \frac{\text{error}_{p,s}}{\min_{s \in \mathcal{S}} \text{error}_{p,s}} \leq \tau \right\} \right|$ given some method $s \in \mathcal{S}$ on tasks $\mathcal{P}$.
- **Software and hardware:** We adopt the free, open-source software *Determined*[2] for experiment management, hyperparameter tuning, and cloud deployment. All experiments are performed on a single p3.2xlarge instance with a 16GB NVIDIA V100 GPU. While evaluation on NAS-Bench-360 indeed assumes access to at least a single V100 GPU, we reiterate that we provide the precomputed NAS-Bench-201 search space for two of our tasks in cases where GPU access is limited. Costs in GPU-hours are in the appendix.

---

[2]https://github.com/determined-ai/determined

### 4.3 Precomputing NAS-Bench-201 on NinaPro and Darcy Flow

The intended goal of NAS-Bench-360 is to evaluate the performance of *NAS search method and search space pairs* on diverse tasks, which precludes the precomputation of all architectures in general due to the lack of a single fixed search space. A complete lack of precomputed architectures would be perhaps limiting for many NAS researchers, who rely on precomputed NAS benchmarks when developing new search methods. In an effort to address this potential limitation, we precompute all architectures in the NAS-Bench-201 [17] search space on two representative tasks in NAS-Bench-360: NinaPro and Darcy Flow. We follow the same experimental procedure as in the original NAS-Bench-201 benchmark [17] to generate the precompute results, except where they vary the number of models trained for each architecture between one and three, we fix the number of trials per architecture to one. Note that NAS-Bench-201 already includes precompute for CIFAR-100, a dataset we include in NAS-Bench-360.

## 5 Analysis

We conclude our presentation of NAS-Bench-360 with three sets of analyses. The first, a performance analysis of NAS methods and fixed baselines across diverse tasks, reveals new insights about the capabilities and robustness of current NAS methods and demonstrates how our benchmark can enable critical next steps in NAS research. In our second analysis, we evaluate claims from the NAS literature originally made using computer vision tasks, and show that they do not generalize to diverse tasks; this demonstrates how NAS research can benefit from our contribution in the future. Finally, we extend an existing analysis of zero-cost proxy methods on diverse tasks that already uses NAS-Bench-360 [52].

### 5.1 Performance across diverse tasks using NAS-Bench-360

As discussed in Section 4.2, we start by considering two practitioners faced with a choice of spending their limited compute on a (possibly tuned) fixed-architecture CNN or trying to find a better architecture using NAS. With this study, we investigate whether modern NAS methods perform well beyond the tasks for which they were designed.

1. A surface-level analysis suggests that under light resource constraints, modern NAS in the form of DARTS (GAEA) is quite robust to a wide variety of tasks: Table 3 shows it is the highest-ranked domain-independent method and attains the most significant improvement over the fixed WRN baseline. The performance profile in Figure 2 (left) also seems favorable, although it is overtaken by tuned WRN at a higher $\tau$-suboptimality. However, a closer look at 2D point tasks in Figure 2 (right) reveals that DARTS is quite poor there, despite its design domain being image classification; in particular, it performs very poorly on NinaPro and FSD50K. Furthermore, on tasks where it performs well, it can still lag behind expert architectures; for example, on Darcy Flow, networks that use FNO [40] or XD-operations [49] do much better. Overall, our results suggest that this practitioner can apply NAS and expect to see some improvement, but also risks catastrophically poor performance (e.g. FSD50K) or not getting truly state-of-the-art results (e.g. Darcy Flow).
2. Under stronger budget constraints, our experiments strongly suggest that a practitioner should simply apply the default WRN to their problem rather than undergo the additional complexity of using DenseNAS, as the latter attains little-to-no improvement over the former in Table 3 and has a usually-worse performance profiles Figure 2. On the other hand, DenseNAS performs well on FSD50K—it outperforms all methods even while DARTS (GAEA) fails.

These first experiments suggest that the modern NAS methods are not always robust to diverse tasks, especially under resource-constrained settings. We believe that NAS-Bench-360's main roles as a future benchmark include developing an understanding of the multi-domain performance of existing approaches and guiding research into better NAS methods. While the latter is beyond the scope of this paper, our additional experiments demonstrate how NAS-Bench-360 facilitates the former.

Notably, several of our results address the question of the relative importance of search space vs. search algorithm. For example, Table 3 shows that on DenseNAS, random search is nearly identical to the more sophisticated weight-sharing scheme of the original paper; the two algorithms' performance profiles are also difficult to distinguish in Figure 2. Furthermore, AMBER—a 1D NAS method whose search space includes larger-kernel convolutions for handling such tasks—does better than GAEA even though it uses an older search algorithm (ENAS). These both suggest that search space design, including the use of a wider variety of operations, may be at least as crucial for success as the search

algorithm. This point is reinforced by example tasks such as Darcy Flow, where architectures with more exotic operations substantially outperform our best results, as discussed earlier.

NAS-Bench-360 also reveals failure points of several methods, not just of general ones that usually perform quite well such as DARTS (GAEA) but also the objective-specific approach Auto-DL, which despite being designed for dense prediction tasks does poorly on all those considered here. Understanding when and why these performance drops happen is critical to developing a more robust NAS that is useful not just on average but in more challenging settings.

## 5.2  Do past NAS-Bench-201 analyses generalize to NAS-Bench-360?

Existing NAS-benches have been widely used for analyses such as (1) comparing performances of different architectures across tasks, (2) quickly evaluating search methods, and (3) investigating design choices that impact performance. In this section we show via the NAS-Bench-201 search space that the conclusions of past analyses cannot be assumed to hold on tasks beyond computer vision.

### 5.2.1  Architecture transferability

We start by using the precomputed results outlined in Section 4.3 to show in Figure 3 the rank of each architecture across different datasets, indexed on the x-axis by its rank on CIFAR-100. This reveals that while architecture rankings are highly correlated on image classification datasets—as pointed out by the authors of the original benchmark [17]—the rankings become uncorrelated when evaluated on a more diversified set of tasks. Therefore, NAS evaluations should be done across domains to verify true generalizability, and NAS-Bench-360 is especially useful for this purpose.

### 5.2.2  Search algorithm performance

Using the precomptued evaluations on two new datasets, we evaluate all ten typical NAS algorithms originally studied on NAS-Bench-201 [17]. The results are shown in the Appendix. With similar wall clock time, the non-weight-sharing NAS algorithms that we evaluate: REINFORCE [54], random search (RS) [5], regularized evolution (REA) [47], BOHB [21], and Hyperband [35] consistently perform well. Our results corroborate the strong performance of non-weight-sharing methods on this search space.

On the other hand, our experiments reveal some important differences for weight-sharing methods. In particular, unlike in past experiments on the NAS-Bench-201 search space, DARTS does not always yield a network of all skip-connection on Darcy Flow, despite this behavior on image classification and NinaPro. Instead, both first-order and second-order DARTS often pick convolution operations and sometimes achieve good performance, although still worse overall than the best non-weight-sharing methods. These results together with the ranking demonstrate that evaluating methods and search spaces on vision tasks alone does not give a full picture of their capabilities and limitations, a problem alleviated by NAS-Bench-360.

### 5.2.3  Operation redundancy

Our final analysis using the NAS-Bench-201 search space is to investigate the conclusions of a more recent study on the redundancy of operations [51]. We find that the operation redundancy phenomenon they outline is task-dependent and does not generalize to tasks beyond the three vision tasks—CIFAR-10, CIFAR-100, and ImageNet16-120—that they study. To conduct our study we follow their procedure to obtain "operation importance" distributions for each operation in the NAS-Bench-201 search space for NinaPro and Darcy Flow; additionally, we reproduce their results on CIFAR-10, CIFAR-100, and ImageNet16-120. *Operation importance* measures the incremental effect of each operation choice in the NAS-Bench-201 search space—1x1 convolutions (c1), 3x3 convolutions (c3), skip connections (skip), and 3x3 average pooling (ap3)—on performance [51]. The original analysis found that the operation importance distributions are roughly similar across the original NAS-Bench-201 computer vision datasets, which we confirm and show in Figure 4. However, we found that the operation importance distributions were drastically different for NinaPro and Darcy Flow, which we also show in Figure 4. Not only are their distributions different from those of the computer vision tasks in the original analysis, but the operation importance distribution for NinaPro differs significantly from that of Darcy Flow. This tells us that *different operations are more useful for different tasks*, and using NAS-Bench-360, we find that we cannot conclude that any of these

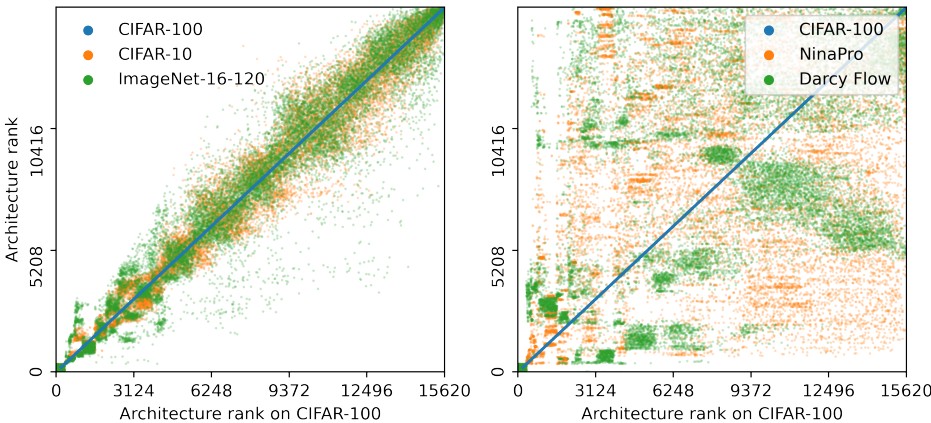

Figure 3: Architecture rankings between computer vision tasks correlate on NAS-Bench-201 [17] (left, sorted by performance on CIFAR-100) but are uncorrelated between CIFAR-100 and two NAS-Bench-360 tasks, NinaPro and Darcy Flow (right).

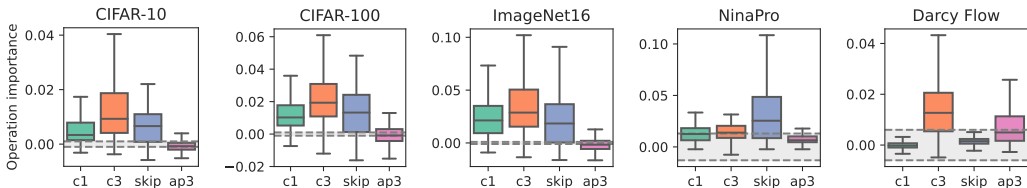

Figure 4: Different operations are important for different tasks. While prior work [51] shows that the operation importance distributions are stable across computer vision tasks—as shown by the high similarity of the three plots on the left—we find that they differ significantly for NinaPro and Darcy Flow.

Table 4: Performance comparison of TE-NAS and GAEA using the DARTS search space on CIFAR-100, Spherical, NinaPro, and Darcy Flow. Lower is better for all metrics.

|        | CIFAR-100 | Spherical | NinaPro | Darcy Flow |
|--------|-----------|-----------|---------|------------|
| TE-NAS | 24.32     | 56.87     | **9.71**| **0.012**  |
| GAEA   | **24.02** | **48.23** | 17.67   | 0.026      |

operations are universally redundant or useful in a given search space across tasks. In other words, using NAS-Bench-360, we find that the original claim that "existing search spaces contain a high degree of redundancy" [51] does not hold when considering diverse tasks beyond computer vision.

## 5.3 Zero-cost proxies on diverse tasks

We conclude with an analysis of TE-NAS [8], a zero-cost proxy inspired by neural tangent kernel (NTK) analysis, on four NAS-Bench-360 tasks. Zero-cost proxies [44, 1] are the subject of a recent direction in NAS research that aims to construct quick-to-evaluate measures of architecture performance without doing any training. Recently, [52] evaluated several zero-cost proxies on tasks from NAS-Bench-360 (Spherical, NinaPro, and Darcy Flow), as well as on TransNAS-Bench-101 [18]. One major weakness of zero-cost proxies that they point out is that zero-cost proxies are not much more computationally efficient than weight-sharing methods, as the total compute cost is still dominated by the evaluation of the searched architecture [52]. For example, this renders TE-NAS in the DARTS search space comparable to GAEA DARTS in terms of computational efficiency. The authors of [52] also point out that the performance of different zero-cost proxies vary considerably across diverse datasets, even subject to the same search space. Performance may be strong on some tasks, but weak on others.

To expand such study of zero-cost proxies, we look at one that [52] do not consider—TE-NAS—and evaluate its performance on the DARTS space using four NAS-Bench-360 tasks: CIFAR-100, Spherical, NinaPro, and Darcy Flow. The results of this evaluation are shown in Table 4. Unlike many other zero-cost-proxies [44], the fact that TE-NAS is constructed from a domain-agnostic NTK analysis rather than experiments makes it a potential candidate for good performance on diverse tasks. However, Table 4 shows that performance does vary considerably across tasks, as observed for other proxies by [52]. In-particular, TE-NAS performs okay on NinaPro and beats all methods in Table 2 on Darcy Flow—where its performance approaches that of the expert-designed FNO [40]—but does poorly on Spherical. This evaluation adds evidence to existing scientific findings already enabled by NAS-Bench-360 [52] and provides additional evidence for the need to evaluate all NAS methods, including zero-cost proxies, on diverse tasks.

## 6  Conclusion

NAS-Bench-360 is a new performance benchmark consisting of ten diverse tasks derived from various fields of research and practice. It is designed for reproducible research on an academic budget that will guide the development of NAS methods and other automated approaches towards more robust performance across different domains. In initial results, we have demonstrated both the need for such a benchmark and the utility of NAS-Bench-360 specifically for developing new search spaces and algorithms. We also provide precompute architectures from the NAS-Bench-201 search space on two of the ten tasks. While the precomputed architectures on these two tasks are useful for analysis on their own, adding more precomputed search spaces and tasks is an area of further improvement. We welcome researchers to use the NAS-Bench-360 tasks to develop new procedures for automating ML.

### Acknowledgments

We thank Maria-Florina Balcan for providing useful feedback. We also thank Hewlett Packard Enterprise for compute resources and the Determined AI open-source community for its support. This work was supported in part by DARPA FA875017C0141, the National Science Foundation grants IIS1705121, IIS1838017, IIS2046613, IIS-2112471, CCF2106707, the American Family Funding Initiative, the Wisconsin Alumni Research Foundation (WARF), an Amazon Web Services Award, a Facebook Faculty Research Award, funding from Booz Allen Hamilton Inc., a Block Center Grant, a Two Sigma Fellowship Award, and a Facebook PhD Fellowship Award. Any opinions, findings and conclusions or recommendations expressed in this material are those of the author(s) and do not necessarily reflect the views of any of these funding agencies.

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
