# OpenReview forum: "NAS-Bench-360: Benchmarking Neural Architecture Search on Diverse Tasks"
_NeurIPS.cc/2022/Track/Datasets_and_Benchmarks — NeurIPS 2022 Datasets and Benchmarks _

### Official Review · Reviewer_bKwR · 2022-07-26
**A good tool for probing NAS robustness**

**Rating:** 6
**Confidence:** 3
**Correctness:** 1. The datasets of this work is a col…

**Strengths:**

1. The question of how well NAS algorithms generalize across tasks is an important one for both practical applications and scientific research.

2. The dataset covers a range of applications, dimensionality, learning objectives, as well as scales.

3. Code, data, models, seeds, and environment are released to facilitate reproduction.

4. Empirical results show many modern NAS methods are not robust as they perform inconsistently across tasks. It identifies an important limitation of many modern NAS methods which may facilitate future research to address them

5. Resource-constrained experiments provide some (potentially counterintuitive) insights for practitioners that they may be better off using a fixed handcrafted domain-specific model.

6. The paper discusses some limitations of the work and provides some insights regarding future improvement.


**Weaknesses:**

1. Perceiver IO needs a certain scale to perform well. While one can argue there is value in testing it at a very small scale, it is not entirely a fair comparison.

2. It is helpful to spell out that the AWS p3.2xlarge has the 16GB version of V100.

3. Natural language processing (NLP) is another important and popular field for modern machine learning which includes NAS research. Is there a good argument for not including NLP tasks?

4. Figure 4 left panel is misaligned


**Additional Feedback:**

Detailed suggestions for improvement can be found in “weakness” and "correctness" sections.

**Clarity:**

The paper is generally well written and easy to follow.


**Documentation:**

Code, data, models, seeds, and environment are released. There is sufficient detail to reproduce results.

**Ethics:**

No ethical concerns are raised as no new datasets are collected.


**Relation To Prior Work:**

The authors provide a good discussion of related work as well as how this work differs from them.


**Summary And Contributions:**

The paper proposes NAS-Bench-360 (NB360), a diverse set of tasks and a collection of metrics for Neural Architecture Search (NAS). NB360 extends beyond popular computer vision benchmarks and some precomputed performance is provided for two tasks. The paper empirically shows inconsistent performance of several modern NAS algorithms across tasks of NB360, demonstrating its usefulness in evaluating the robustness of NAS methods.

---

> ### Author Response · Authors · 2022-08-11
> **Reply to Reviewer bKwR**
>
> We thank the reviewer for the insightful comments, questions, and for acknowledging the importance of our work. We have responded to each of your comments below.
>
> > "Perceiver IO needs a certain scale to perform well. While one can argue there is value in testing it at a very small scale, it is not entirely a fair comparison."
>
> We thank the reviewer for raising this point about Perceiver IO to which we largely agree. Indeed, it is unsurprising that Perceiver IO underperforms in lower-data settings. From our perspective, given the success of transformer-based models on a variety of tasks and the broad applicability of Perceiver IO, we found it to be a necessary point of comparison. That said, we have updated the draft to clearly point out that training Perceiver IO from scratch in these smaller-scale data settings might not be the best fit for it.
>
> > "It is helpful to spell out that the AWS p3.2xlarge has the 16GB version of V100."
>
> We thank the reviewer for pointing this out. We have updated the paper with this clarification.
>
> > "Natural language processing (NLP) is another important and popular field for modern machine learning which includes NAS research. Is there a good argument for not including NLP tasks?"
>
> Our aim is to evaluate diverse and understudied tasks. Since NLP has been well-studied both within the NAS community and within the broader context of machine learning, we chose to exclude such tasks. We note that prior work already benchmarks NAS on NLP tasks [2], but prior work on NAS evaluation for other 1d tasks is quite limited, so we view this contribution as an interesting new avenue for NAS evaluation.
>
> [2] https://arxiv.org/pdf/2006.07116.pdf
>
> > "Figure 4 left panel is misaligned"
>
> We’ve fixed the alignment of Figure 4. Thank you for pointing this out!
>
> > "The computation restriction of requiring a model to fit on a single V100 GPU, although could be justified, is not justified in the paper. On one hand some labs can not afford V100 GPUs, on the other hand many architectures (like Perceiver IO in the paper) really require a certain scale to perform well which needs multi-GPU training. In that regard, it is unsurprising to see Perceiver IO underperform in the paper."
>
> We thank the reviewer for pointing this out. We have updated the paper with justification along the following lines: in many cases, access to at least a V100 GPU is a reasonable ask in terms of compute resources, however, we provide the precomputed NAS-Bench-201 search space for two of our tasks in cases in which researchers do not have access to GPU resources.

---

### Official Review · Reviewer_TZcQ · 2022-07-26
**An important step towards generalizing NAS beyond Computer Vision**

**Rating:** 6
**Confidence:** 3
**Clarity:** The paper is well-written, easy to fo…

**Strengths:**

- Correctly identifies and addresses an issue with existing NAS research, namely, overfitting on a handful of CV tasks.
- Sensible and valid analyses of the presented work and the proposed hypotheses.
- Open-source code has been made available for facilitating reproducibility.
- Benchmark/dataset hosting appears reliable and the licenses are open.
- The paper has been written well and is easy to understand.

**Weaknesses:**

- There are some incomplete and missing analyses in the paper. Unless I somehow missed it, the analyses mentioned in Section 5.2.2 that should have been present in the appendix were absent.
- There is room for improvement by conducting an internal validation of the diversity of the chosen tasks, thereby justifying their applicability to the experiments conducted here, i.e. to testing how badly NAS algorithms generalize. The "expert" models mentioned in appendix D would have made for a very decent baseline for demonstrating this, by illustrating the drops in performance in an "expert" model when it is used for a task that it has not been designed for. I must admit, however, that I am unaware of the computational costs involved in conducting such an analysis.
- The authors have a very rich dataset at hand which could have been used for a much more interesting and in-depth statistical analysis than has been presented, missing out on a lot of potential for the paper, going beyond comparing only the final performances of NAS algorithms across tasks but also including an analysis of their behaviours in multiple dimensions. (See "Additional Feedback" for an example)

**Additional Feedback:**

1. The claims made in section 5.2.2 hinge on the presence of an accompanying analysis in the appendix, which I was unable to locate.
2. Have you considered using a log-scale on at least one of the axes in Figure 2 for the sake of visual clarity?
3. Line 180-181: The last sentence here probably needs to be rephrased or is missing a phrase, that "since" in there makes no sense to me.

I am aware that the following apply only to the Appendix, and hence am including this here only as possible "bonus points":
- Appendix F looks very much incomplete.
- Possible extra statistical analysis: As tables 8 and 9 demonstrate, there exists both a significant disparity in the number of FLOPS for the models being searched by different algorithms in different search spaces and for different tasks. This warrants a possibly very interesting statistical analysis of the impact this has on the final performance for a given task as well as changes in behaviour across tasks. How much disparity does exist in the size/computational capacity (FLOPS) of the models searched by the same search algorithm/search space across tasks? How correlated are model capacity (FLOPS) and model performance (figure 5 only show model size vs performance)? I imagine the results of such an analysis on behavioural disparities across tasks would actually also fit and complement the main paper very well. Moreover, you should technically already have all the data needed for such an analysis.

**Correctness:**

To the best of my understanding, the evaluations and experiments carried out in the paper are justified and valid.

**Documentation:**

The data collection and processing has been documented, the licenses have been clearly mentioned and appropriate URLs are available. The code repository has also been documented well.

**Ethics:**

To the best of my understanding and knowledge, there are no ethical concerns that need to be raised.

**Relation To Prior Work:**

A related works section has been presented with appropriate contextualization of the presented work.

**Summary And Contributions:**

This work raises a concern about recent NAS research overfitting on Computer Vision (CV) tasks and attempts to alleviate the issue by providing NAS researchers a benchmark that assesses their work's viability for tasks beyond widely popular CV tasks and instead for tasks generally conducive to the use of Convolutional Neural Networks instead. The primary contribution of the authors is a collection of tasks, including datasets and performance metrics, which they have named "NAS-Bench-360", that aims to provide a standardized baseline for comparison of any NAS algorithm's performance across multiple task domains. A secondary component of their work is a dataset of pre-computed neural network configuration performances for two of the 10 tasks included NAS-Bench-360, in the same vein as and supplemental to that provided by NAS-Bench-201 on 3 CV tasks.

The authors use their proposed benchmark to evaluate a number of NAS search spaces and algorithms in order to demonstrate that many existing state-of-the-art works in the field fail to generalize well to tasks other than CV. They also present an analysis of the viability of using NAS for researchers on constrained budgets working on specific research tasks that could potentially benefit from employing NAS but do not necessarily need to. Currently, some further analyses have been mentioned in the paper but are missing from the appendix.

Overall, a well written, coherent paper which makes a valuable contribution to the scientific community. Once the missing analyses have been added in, I would be happy to increase the rating of the paper.

---

> ### Author Response · Authors · 2022-08-11
> **Reply to Reviewer TZcQ**
>
> We thank the reviewer for the detailed and thorough review, as well as for their acknowledgement of the importance of our work! We have addressed your points below.
>
> > "There are some incomplete and missing analyses in the paper. Unless I somehow missed it, the analyses mentioned in Section 5.2.2 that should have been present in the appendix were absent."
>
> These results are shown in Table 5 (Appendix) and in Appendix A.
>
> > "There is room for improvement by conducting an internal validation of the diversity of the chosen tasks, thereby justifying their applicability to the experiments conducted here, i.e. to testing how badly NAS algorithms generalize. The "expert" models mentioned in appendix D would have made for a very decent baseline for demonstrating this, by illustrating the drops in performance in an "expert" model when it is used for a task that it has not been designed for. I must admit, however, that I am unaware of the computational costs involved in conducting such an analysis."
>
> This is an excellent point, and we thank the reviewer for raising it. We are currently looking into this. Additionally we would like to point to some of our existing results which are relevant to this point: we evaluate the Wide ResNet architecture as a baseline on all of our tasks. In Table 2, our results show significant performance drops of this baseline architecture from CIFAR-100 to Spherical CIFAR-100 and FSD50K.
>
> > "Have you considered using a log-scale on at least one of the axes in Figure 2 for the sake of visual clarity?"
>
> We thank the reviewer for the suggestion. We have updated Figure 2 accordingly, however, we note that since the values on both axes are small, log scale does not make a significant visual difference on either axis. Note that these plots have changed very slightly as we have fixed a minor plotting bug.
>
> > "Line 180-181: The last sentence here probably needs to be rephrased or is missing a phrase, that "since" in there makes no sense to me."
>
> We have fixed this. Thank you for pointing this out.
>
> > "Possible extra statistical analysis: As tables 8 and 9 demonstrate, there exists both a significant disparity in the number of FLOPS for the models being searched by different algorithms in different search spaces and for different tasks. This warrants a possibly very interesting statistical analysis of the impact this has on the final performance for a given task as well as changes in behaviour across tasks. How much disparity does exist in the size/computational capacity (FLOPS) of the models searched by the same search algorithm/search space across tasks? How correlated are model capacity (FLOPS) and model performance (figure 5 only show model size vs performance)? I imagine the results of such an analysis on behavioural disparities across tasks would actually also fit and complement the main paper very well. Moreover, you should technically already have all the data needed for such an analysis."
>
> We thank the reviewer for the detailed and insightful suggestion. We are currently looking into this.

---

> > ### Author Response · Authors · 2022-08-22
> > **Experimental results**
> >
> > To address your suggestions and questions, we have conducted two additional sets of experiments and present the results here. We will also add these results to the appendix of our submission.
> >
> > > "There is room for improvement by conducting an internal validation of the diversity of the chosen tasks, thereby justifying their applicability to the experiments conducted here, i.e. to testing how badly NAS algorithms generalize. The "expert" models mentioned in appendix D would have made for a very decent baseline for demonstrating this, by illustrating the drops in performance in an "expert" model when it is used for a task that it has not been designed for. I must admit, however, that I am unaware of the computational costs involved in conducting such an analysis."
> >
> > We thank the reviewer for the suggested experiment. While computationally costly to evaluate every expert model/task pair across multiple seeds, we have indeed confirmed this drop in performance using the expert FNO architecture on our three dense tasks: Darcy Flow, PSICOV, and Cosmic. FNO was designed for Darcy Flow and achieves the best performance on this task across all methods.  In contrast, we find that FNO performance is considerably worse on the other two dense tasks, PSICOV and Cosmic (as shown below).
> >
> >
> > **Performance (lower is better) of FNO across dense tasks**
> > |                     | Darcy Flow  | PSICOV     | Cosmic      |
> > |---------------------|-------------|------------|-------------|
> > | Expert (Darcy Flow) | **0.008±0.001** | 4.43±0.055 | 54.57±0.030 |
> > | Expert              | **0.008±0.001** | **3.35±0.140** | **25.29±1.440** |
> >
> >
> >
> > > "How correlated are model capacity (FLOPS) and model performance (figure 5 only show model size vs performance)?"
> >
> > We thank the reviewer for suggesting this analysis. We have conducted a version of this suggested analysis using the 15,625 precomputed NAS-Bench-201 architectures trained on CIFAR-100, NinaPro, and Darcy Flow. We find that FLOPS and performance (measured using negative loss) exhibit minimal correlation, especially among the more performant architectures in the search space. We compute this correlation using the top 5%, 25%, 50%, and among all of the architectures in the search space (i.e., 100%). We report these results in the table below. Remarkably, we see (small) negative correlation between FLOPS and model performance in many cases.
> >
> > **Correlation between FLOPS and performance**
> > | % of top architectures | CIFAR-100 | NinaPro | Darcy Flow |
> > |------------------------|-----------|---------|------------|
> > | 5%                     | -0.2744   | -0.0443 | 0.3928     |
> > | 25%                    | -0.1027   | -0.0812 | 0.3026     |
> > | 50%                    | -0.0230   | -0.0608 | 0.4593     |
> > | 100%                   | 0.2838    | 0.0119  | 0.4668     |

---

### Official Review · Reviewer_bqEm · 2022-07-28
**An ordinary research**

**Rating:** 5
**Confidence:** 5
**Correctness:** Yes
**Clarity:** Yes

**Strengths:**

1. This article is very well organized and written.
2. The article gives a more detailed analysis and experiments.

**Weaknesses:**

The biggest problem idea is similar to Trans-bench by Huawei in CVPR[1]：
1. The multitasking selection of this work seems less scientific compared to Trans-Bench.
2. Lack of evaluation of One-shot NAS, zero-shot NAS.
3. This work has actually been proposed for some time, but no other NAS work has been reviewed on it.

[1]  TransNAS-Bench-101: Improving Transferability and Generalizability of Cross-Task Neural Architecture Search(TransNAS-Bench-101)

**Additional Feedback:**

None

**Documentation:**

Yes

**Ethics:**

Yes

**Relation To Prior Work:**

No

**Summary And Contributions:**

This work presents a benchmark for evaluating NAS methods on different tasks. This work has done more evaluation and analysis. However similar work already exists and some of their tasks are more reasonably selected.

---

> ### Author Response · Authors · 2022-08-11
> **Reply to Reviewer bqEm**
>
> Thank you for your feedback. We respectfully but strongly disagree with the weaknesses you discuss. In particular, you raised concerns related to the connections to prior work, our experimental evaluation, and the current level of adoption of NAS-Bench-360. We believe that these claims are either incorrect or misplaced, as we describe below.
>
> > "The biggest problem idea is similar to Trans-bench by Huawei in CVPR"
>
>
> We disagree with this claim. Three key differences between our work and TransNAS-Bench-101 are as follows:
> 1. TransNAS-Bench-101 comprises only computer vision tasks, while our primary objective is to move NAS evaluation beyond computer vision tasks. Along this axis, the goals of TransNAS-Bench-101 are actually the opposite of ours in NAS-Bench-360.
> 2. TransNAS-Bench-101 comprises seven distinct learning objectives over a single vision dataset, whereas NAS-Bench-360 comprises 10 distinct datasets from different domains.
> 3. TransNAS-Bench-101 aims to evaluate architecture transfer, which is not a goal of our work. As such, NAS-Bench-360 should not be evaluated along this axis as it is not directly designed for this problem.
>
> > "The multitasking selection of this work seems less scientific compared to Trans-Bench."
>
> Again, we disagree with this claim, and request that the reviewer provide some justification behind this opinion. From our perspective, the types of problems and task desiderata between the two benchmarks are simply very different and are not directly comparable. We selected tasks to be diverse along a variety of axes: task domains beyond computer vision, dataset sizes, input dimensionalities, output dimensionalities, learning objectives, and evaluation criteria. In contrast, TransNAS-Bench-101 uses a single image dataset with a variety of learning objectives and evaluation criteria.
>
> > "Lack of evaluation of One-shot NAS, zero-shot NAS."
>
>
> This is an inaccurate claim. Please refer to section 5 of our paper. We evaluate several one-shot weight-sharing methods as well as zero-cost proxies (i.e., zero-shot NAS). These evaluations comprise a significant portion of our main results.
>
> > "This work has actually been proposed for some time, but no other NAS work has been reviewed on it."
>
> We believe that the **current** level of adoption of a piece of work, both as it pertains to ours and to research more broadly, is not a fair axis of evaluation for publication. Moreover, few would disagree that publication increases visibility and the potential for adoption.

---

> > ### Comment · Reviewer_bqEm · 2022-08-15
> > **I am disappointed with the author's attitude towards the paper.**
> >
> > The author's response did not remove my doubts.
> > 1. I don't think benchmarking tasks other than computer vision is a great contribution: as we all know, computer vision is a more difficult task than other tasks such as language, text, signal processing, etc. Do you think it is a meaningful contribution to the community to provide some simpler tasks other than vision?
> > 2. this benchmark is too simple and does not provide a valid assessment of the NAS approach.
> > This data is mostly classification tasks on simple datasets. But the more difficult ones include regression, and reconstruction tasks are not included.
> > 3. very inadequate evaluation of NAS methods on this dataset: although this work provides a few NAS methods, this is not enough, please provide more methods to test: e.g. One-Shot NAS: SPOS, OFA, BigNAS, FairNAS, etc; Zero-NAS: FLOPs. NWOT, epe-nas，fisher，flops，grad-norm，grasp， l2-norm，jacov ，nwot，params ，snip， synflow ，zen-score, etc;
> > 4. please provide the results of random NAS search with FLOPs as an indicator, if this result is already better than Darts method, then this benchmark is useless.
> > 5. this work has been rejected several times, but we don't see much improvement in the revisions.

---

> > > ### Author Response · Authors · 2022-08-17
> > > **Response**
> > >
> > > We again strongly disagree with your views. Please see below for further clarifications.
> > > 1. We do not believe most of the machine learning or even computer vision community would share your belief that “computer vision is a more difficult task than other tasks such as language, text, signal processing, etc.”
> > > 2. This is incorrect. According to the relevant measure of difficulty—whether modern NAS methods perform well—our evaluation shows that most of the tasks are indeed difficult. NAS methods fail to beat human-designed architectures on five of the ten tasks, and on three more at least one NAS method performs catastrophically poorly.
> > > 3. Our level of evaluation is comparable to past benchmarks in the field. Excluding variants of the same method, we evaluate only one less one-shot method than NAS-Bench-201 (Dong & Yang, 2020), while TransNAS-Bench-101 does not seem to evaluate any one-shot methods at all; both also evaluate on fewer tasks. We believe there should be a strong justification when asking to evaluate any additional method; in particular, we have not seen most of the methods you suggest in the original evaluations of any existing NAS-Bench. Furthermore, many of your zero-cost proxy suggestions *have* been evaluated on a subset of our benchmark in subsequent work and were found to have very mixed performance (White et al., 2022).
> > > 4. We disagree with the premise of your request: the poor performance of DARTS (or any other method) relative to another method, even FLOPs, does *not* imply a benchmark is “useless.” For example, it is well-known that DARTS converges to networks of all skip-connections on NAS-Bench-201 (Dong & Yang, 2020), something which FLOPs would not do; this does not mean NAS-Bench-201 is “useless.” FLOPs have also been found to perform well relative to other zero-cost proxies on several tasks from TransNAS-Bench-101 (White et al., 2022, Table 3); this does not mean TransNAS-Bench-101 is “useless.” In any case, White at al. (2022, Table 2) found that FLOPs were *negatively* correlated with the performance of DARTS architectures on four of the five NAS-Bench-360 tasks they tested; in-fact, the one task where there *was* positive correlation was CIFAR-100.
> > > 5. This is incorrect: among other improvements, our current submission augments our original draft with precomputed performance of 15,625 architectures on two of the tasks, an entire new section (Section 5) of additional experimental analysis, and evaluations of non-NAS baselines on all tasks.
> > >
> > > &nbsp;
> > > **References:**
> > > 1. Dong,Yang. *NAS-Bench-201: Extending the Scope of Reproducible Neural Architecture Search*. ICLR 2020.
> > > 2. White, Khodak, Tu, Shah, Bubeck, Dey. *A Deeper Look at Zero-Cost Proxies for Lightweight NAS*. ICLR 2022 Blog Track.

---

> > > > ### Comment · Reviewer_bqEm · 2022-08-25
> > > > **Thank you for the response.**
> > > >
> > > > Thank you for your reply. I still think your response and the current version does not satisfy me. I keep my score.

---

### Official Review · Reviewer_Ye1M · 2022-07-28
**Well motivated work**

**Rating:** 6
**Confidence:** 4
**Clarity:** The paper is well structured and clea…

**Strengths:**

- A diverse collection of tasks with scripts provided to evaluate and reproduce results of different NAS algorithms and search spaces.
- Comprehensive experimental results showing the shortfall of exisiting algorithms and search space on generalization.

**Weaknesses:**

- As the authors already mentioned, this 'benchmark' is an organization of tasks such that users can evalute different NAS algorithms and search spaces. Since it is not a pre-computed tabular benchmark, I wonder if this work can be called a benchmark.
- Also I am interested to see if the authors have provided an API or standardization to allow people to contribute new search algorithms, search spaces and precomputed data.


**Additional Feedback:**

- In the experiment, how do you define a good set of hyperparameters for the constrained setting?
- It is expected that a search space for one task does not perform well on another task. Are there any other search spaces specifically designed for those non-computer vision tasks?
- Section 5 is referencing Table 5, should it be Table 4?

**Correctness:**

- To the best of my knowledge, the dataset is constructed in a sound way, together with well structured evaluation methods.

**Documentation:**

Some sections in the Appendix are missing contents, e.g. F3, F4, F5.

**Ethics:**

No concern as far as I am aware.

**Relation To Prior Work:**

It is clearly discussed.

**Summary And Contributions:**

This paper presents a benchmark suite which is a collection of datasets / tasks to enable evaluate of different search spaces and search algorithms. The datasets also include precomputed performance of NAS-Bench-201 architectures on two of the tasks.
Based on the tasks collected, the paper presents experimental results of several NAS algorithms on multiple search spaces and tasks. The authors conclude that the analysis by previous papers on computer vision tasks do not generalize to diverse tasks. In addition, the zero-cost proxies perform inconsistently as well.

---

> ### Author Response · Authors · 2022-08-11
> **Reply to Reviewer Ye1M**
>
> We thank the reviewer for the helpful questions as well as for the praise of the motivation of our work. We have responded to each of them below.
>
> > "As the authors already mentioned, this 'benchmark' is an organization of tasks such that users can evaluate different NAS algorithms and search spaces. Since it is not a pre-computed tabular benchmark, I wonder if this work can be called a benchmark."
>
> We would like to emphasize that while we (necessarily) do not precompute every architecture in every search space considered in our evaluation, NAS-Bench-360 is still nonetheless a benchmark of NAS methods. Please see point 1 of the general response for a more detailed discussion.
>
> > "Also I am interested to see if the authors have provided an API or standardization to allow people to contribute new search algorithms, search spaces and precomputed data."
>
> Yes, this is precisely the motivation behind our work! We intend for this work to be used by the NAS community to develop new methods and we have provided all code necessary to do so. We have released all benchmarking and analysis code, including data loaders, splits, evaluation code, and the NAS-Bench-201 API for the precomputed portion of our benchmark.
>
> > "Some sections in the Appendix are missing contents, e.g. F3, F4, F5."
>
> We apologize for the misleading formatting for these subsections (which originally contained only figures). We have fixed this. Thank you for pointing this out!
>
> > "In the experiment, how do you define a good set of hyperparameters for the constrained setting?"
>
> In the constrained setting, we use the default set of hyperparameters for Wide ResNet (https://github.com/meliketoy/wide-resnet.pytorch) and DenseNAS (https://github.com/JaminFong/DenseNAS), aside from the batch size which is adjusted per task.
>
> > "It is expected that a search space for one task does not perform well on another task. Are there any other search spaces specifically designed for those non-computer vision tasks?"
>
> We study domain-specific search spaces through the inclusion of Auto-DL, designed for 2D dense prediction, and AMBER, designed for 1D prediction, in NAS-Bench-360 evaluations. These search spaces are explained in more detail in Appendix C, under "Domain-specific NAS Baselines." Results from Table 2 show that even these specialized search spaces do not perform well within their own domain.
>
> > "Section 5 is referencing Table 5, should it be Table 4?"
>
> This has been fixed. Thank you for pointing this out!

---

> > ### Comment · Reviewer_Ye1M · 2022-08-24
> > **Thank you for the response.**
> >
> > I can see the motivation of this work, and it could potentially raise the awareness about the limitation of existing NAS benchmarking. Because of this I am happy to see this paper accepted. However, as the other reviewers mentioned, this paper can be improved by having more statistical analysis and inclusion of NLP tasks. I keep my score in this case.

---

> > > ### Author Response · Authors · 2022-08-29
> > > **Additional statistical analysis**
> > >
> > > Thank you for the reply. We would like to note that we have added an additional statistical analysis—[a correlation analysis between FLOPS and model performance](https://openreview.net/forum?id=xUXTbq6gWsB&noteId=TsRq8BTozHw)—in response to Reviewer TZcQ.

---

### Official Review · Reviewer_no5b · 2022-07-29
**Important idea with missing insights**

**Rating:** 5
**Confidence:** 4
**Clarity:** This paper is clearly and well written.

**Strengths:**

+ This paper tackles an important fact, that NAS methods should be evaluated on different domains to indeed show superiority of a proposed NAS method.
+ The experiments underline the importance of this tasks and the claimed contribution.
+ Most of the selected tasks are reasonable and sufficiently different for this evaluations
+ The paper is well written and clearly structured
+ The provided code and URL are clear.

**Weaknesses:**

- The main concern, is that this paper claims to introduce a benchmark, whereas this paper mainly introduces an evaluation of different NAS methods on different task.
-  A benchmark should provide detail information about the found architectures and their performance in their provided documentation. In the best case also the learning curves or the weights for future research.
- Some of the considered datasets only have a small size, especially the datasets, which are used for the NAS-Bench-201 evaluations.


**Additional Feedback:**

As stated in the weakness part some of the considered datasets are small. Does this small dataset size affect the performance of the NAS-Bench-201 architectures in any way?
Also due to the different NAS methods also different search spaces are compared with each other. How do the actual underlying predefined search spaces influce the overall ability to transfer methods to other domain tasks? Do some of the NAS methods perform better, because of the search method and optimizer or because of the search space? That is not clear to me.
Where are the performances of networks using FNO or XD-operations in the paper?
In addition it is not clear to me how XGBoost is included? What is the underlying architecture search space on which XGBoost is applied to as a regression task?

Given the weaknesses and the additional feedback, I tend to reject this paper.
However, I hope that the authors can address my concersn in the discussion phase.


**Correctness:**

The construction of this paper is clear and reasonable and the made claims are correct given the shown experiments.
However some of the claims (see Additional Feedback) are not clear to me.


**Documentation:**

This paper provides sufficient detail to support reproducibility as well as information regarding data collection and organization.

**Ethics:**

No.

**Relation To Prior Work:**

This paper includes all important related work and clearly states the difference to prior work and previous contributions.

**Summary And Contributions:**

This paper introdcues NAS-Bench-360, which evaluates different architectures and NAS methods on in total 10 different tasks. This evaluation shows, that different NAS methods, which perform very well on computer vision tasks do not transfer with a high performance to other domains. This paper also evaluates the architectures from the NAS-Bench-201 search space on 2 diverse tasks, showing that these architectures do not generalize well. Lastly, this paper compares TE-NAS, a zero-shot search method, with GAEA on several tasks, resulting in varied performance over these considered diverse tasks.

---

> ### Author Response · Authors · 2022-08-11
> **Reply to Reviewer no5b**
>
> We thank the reviewer for the insightful questions and we appreciate the reviewer’s acknowledgement of the importance of this research direction. We have itemized our responses to your points below.
>
> > "The main concern is that this paper claims to introduce a benchmark, whereas this paper mainly introduces an evaluation of different NAS methods on different tasks."
>
> We would like to emphasize that while we (necessarily) do not precompute every architecture in every search space considered in our evaluation, NAS-Bench-360 is still nonetheless a benchmark of NAS methods. Please see point 1 of the general response for a more detailed discussion.
>
> > "A benchmark should provide detailed information about the found architectures and their performance in their provided documentation. In the best case also the learning curves or the weights for future research."
>
> We agree with this. This metadata for the precomputed NAS-Bench-201 architectures for Darcy Flow and NinaPro is available via our easy-to-use benchmark API that contains the validation accuracy/loss for every architecture after every epoch (learning curve) as well as model weights for every architecture. We have updated the instructions for benchmark usage in our code repository. Furthermore, for our broader evaluation in which we do not precompute all architectures in a fixed search space, we provide both the random seeds to reproduce these results and the architecture specifications of the resulting models.
>
> > "Some of the considered datasets only have a small size, especially the datasets, which are used for the NAS-Bench-201 evaluations."
>
> This is again by design. The datasets in NAS-Bench-360 comprise a wide range of sizes – this is by design so as to also evaluate task diversity in terms of dataset size. The smallest dataset has 1,100 samples while the largest has 1,000,000 samples.
>
> > "As stated in the weakness part some of the considered datasets are small. Does this small dataset size affect the performance of the NAS-Bench-201 architectures in any way?"
>
> We note that the lower parameter counts of the architectures in NAS-Bench-201 actually helps in these lower-data settings. Quantitatively, we find that 160 architectures from the NAS-Bench-201 search space outperform Wide ResNet on the smaller NinaPro DB5 dataset whereas none of the NAS-Bench-201 architectures outperformed Wide ResNet on the larger CIFAR-100 dataset.
>
> > "Also due to the different NAS methods also different search spaces are compared with each other. How do the actual underlying predefined search spaces influence the overall ability to transfer methods to other domain tasks? Do some of the NAS methods perform better, because of the search method and optimizer or because of the search space? That is not clear to me."
>
> We thank the reviewer for raising these questions. We agree that our analysis focuses on evaluating NAS methods as search space/search algorithm pairs, and as we discussed in the general response, we made this decision to facilitate the study of NAS for diverse tasks. While a more in-depth evaluation of search spaces themselves across diverse tasks is not the main focus of our analysis, we agree that this is an interesting question. Indeed, empowering the broader research community to ask a wide range of scientific questions (including this one) is precisely the motivation behind our creation of NAS-Bench-360.
>
> > "Where are the performances of networks using FNO or XD-operations in the paper?"
>
> We evaluate FNO on Darcy Flow (see Appendix D). However, since FNO is designed for PDE problems as opposed to diverse tasks, we do not evaluate FNO-based architectures on other tasks. We furthermore exclude XD-operations from our evaluation, as XD-operations are too costly to evaluate on all of our tasks, as noted in [1].
>
> [1] https://arxiv.org/pdf/2204.07554.pdf
>
> > "In addition it is not clear to me how XGBoost is included? What is the underlying architecture search space on which XGBoost is applied to as a regression task?"
>
> We take the position that NAS methods should be evaluated against not only other NAS methods, but against other strong baselines on diverse tasks. XGBoost is known to perform well on diverse tasks, so we include it as a fixed non-NAS (and non-deep learning) baseline in our evaluation. We note that XGBoost is generic enough to apply to all of our tasks without modification.

---

> > ### Comment · Reviewer_no5b · 2022-08-29
> > **Thank you for the response**
> >
> > Thank your for the response.
> > I understand the motivation for this paper, nonetheless, I still think that providing information about only a subset of tasks should not be the goal for a benchmark for reproducible research. In addition, I agree with the statements by other reviewers that more statistical analysis and more diverse tasks (NLP) would improve this paper. Thus, I keep my score.

---

> > > ### Author Response · Authors · 2022-08-29
> > > **Additional statistical analysis**
> > >
> > > We appreciate your reply. We would like to note that we have added an additional statistical analysis—[a correlation analysis between FLOPS and model performance](https://openreview.net/forum?id=xUXTbq6gWsB&noteId=TsRq8BTozHw)—in response to Reviewer TZcQ. Furthermore, we note that the reviewers who have asked for statistical analyses and NLP tasks (TZcQ and bKwR) have given our submission a score that tends toward accepting the paper (i.e., score of 6).

---

### Author Response · Authors · 2022-08-11
**General Response**

We thank the reviewers for the helpful feedback, which has significantly improved the presentation of our results. We would like to clarify the following two common points that came up across several reviews.

1. We would like to reiterate a major motivation of our work: **NAS for diverse tasks inherently requires the evaluation of a diverse set of search spaces as well as search methods.** This means that the traditional model of benchmarking in the NAS community, NAS-Benches with precomputed evaluation of a single search space, is insufficient. While in this sense, our work necessarily diverges from traditional NAS-Benches, NAS-Bench-360 is still nonetheless a benchmark of NAS methods, although we consider a NAS method to be a (search space, search algorithm) pair. We furthermore view this as a primary contribution of our work, as it both challenges and expands the standard understanding of what it means for a NAS benchmark to be a NAS-Bench. Nonetheless, we understand the value of precomputed tabular NAS-Benches, so we still provide all precomputed architectures from the NAS-Bench-201 search space on a subset (2) of our tasks so as to enable researchers who have limited access to GPUs, though we emphasize that the precomputed evaluation is limited to only one search space (NAS-Bench-201).
2. **We would like to clarify that our tasks were selected to be diverse along several dimensions:** dataset size, problem domain, input/output dimensionality, learning objective, and evaluation criterion. In particular, we have selected our tasks to represent a wide range of dataset sizes while simultaneously maintaining reasonable efficiency – as with other NAS-Benches, our intention is to enable researchers with a non-industry budget to evaluate their methods using our benchmark.

---

### Comment · Reviewer_bqEm · 2022-08-15
**The NAS community needs new breakthrough approaches and difficult benchmarks, and this ordinary work is not enough.**

Dear reviewers, I am very curious about the positive opinion of most of you about this very common work (rejected many times).
The NAS community hasn't had a breakthrough for a long time, and some benchmarks NAS101/201/301 and so on have been plentiful, and work similar to this one exists ([1] TransNAS-Bench-101: Improving Transferability and Generalizability of Cross-Task Neural Architecture Search(TransNAS-Bench-101)).
I think this work is very common and needs to be promoted for the following reasons：

1. I don't think benchmarking tasks other than computer vision is a great contribution: as we all know, computer vision is a more difficult task than other tasks such as language, text, signal processing, etc. Deep learning methods have been doing very well on these signal processing tasks for a long time. Evaluating NAS methods on these simple tasks is not convincing and does not have much implications.
2. this benchmark is too simple and does not provide a valid assessment of the NAS approach. This data is mostly classification tasks on simple datasets. But the more difficult ones include regression, and reconstruction tasks are not included.
3. very inadequate evaluation of NAS methods on this dataset: although this work provides a few NAS methods, this is not enough, please provide more methods to test: e.g. One-Shot NAS: SPOS, OFA, BigNAS, FairNAS, etc; Zero-NAS: FLOPs. NWOT, epe-nas，fisher，flops，grad-norm，grasp， l2-norm，jacov ，nwot，params ，snip， synflow ，zen-score, etc;
4. please provide the results of random NAS search with FLOPs as an indicator, if this result is already better than Darts method, then this benchmark is useless.
5. this work has been rejected several times, but we don't see much improvement in the revisions.

As a researcher in the NAS community, I am very eager for breakthrough work to appear. Ordinary work like this is not good enough to help the development of NAS.

---

> ### Author Response · Authors · 2022-08-17
> **Please see our response below**
>
> We strongly disagree with the views expressed by this reviewer, and have responded to the points directly addressed to us in the discussion of the original review by BqEm below. Please refer to [this response](https://openreview.net/forum?id=xUXTbq6gWsB&noteId=RLqckoaJ0_n) for clarifications on the above.

---

### Author Response · Authors · 2022-08-24
**Feedback on responses**

We thank the reviewers for their reviews---we have responded to your comments. As this discussion period draws to a close, we would like to solicit feedback on our responses and any additional feedback you might have regarding our submission. Thank you!

---

### Meta-Review · Area_Chair_GWRr · 2022-09-09

**Recommendation:** Accept
**Confidence:** 3

**Metareview:**

NAS-Bench-360 enables the evaluation of NAS methods across a diverse set of tasks and domains and the author's experiments demonstrate that NAS methods are not generally robust across all domain/tasks.

The authors also make a strong argument that a NAS benchmark should be a combination of a search space and a search algorithm, challenging pre-conceived notions of what a benchmark is for this research community.  (Existing pre-computed tabular benchmarks do not necessarily allow for innovation of the search space.)

Overall, the benchmark and evaluations behind NAS-Bench-360 will benefit the broader research community and are significant improvement over existing work. The authors should consider adding NLP tasks to strengthen the paper.

---

### Decision · Program_Chairs · 2022-09-16

Accept